# OpenReview forum: "BARRED: Synthetic Training of Custom Policy Guardrails via Asymmetric Debate"
_ICML.cc/2026/Conference — ICML 2026 regular_

### Official Review · Reviewer_pgTe · 2026-02-26

**Soundness:** 3
**Presentation:** 2
**Significance:** 2
**Originality:** 2
**Overall Recommendation:** 4
**Confidence:** 3

**Summary:**

This paper proposed BARRED: a multi-agent debate-based data synthesis framework for training custom policy guardrails. The pipeline first decomposes the task into a few-dimensional instances, then sample data points from these instantiations. Afterwards, through iterative multi-agent debate, the pipeline selects the datapoints that all the judges reach an agreement in the end. Experiment results showed that the guardrail model trained on BARRED has significantly better performance compared with generate guardrail model.

**Compliance With Llm Reviewing Policy:**

Affirmed.

**Ethical Review Concerns:**

Some personally identifiable information is leaked in the draft. In line 108, the author pointed out the dataset link to a public huggingface repo, which may indicate the author's affiliation.

**Ethical Review Flag:**

Flag this paper for an ethics review.

**Ethics Expertise Needed:**

["Privacy and Security (e.g., personally identifiable information)"]

**Final Justification:**

Most of my concerns have been addressed. Overall, the framework proposed is novel, and the experiment is comprehensive. Though the presentation still needs improvement, e.g., more experiment setup explanation, more detailed table/figure captions are desired.

**Key Questions For Authors:**

I have listed my questions in the limitations section.

**Limitations:**

yes

**Strengths And Weaknesses:**

### Strengths
1. The paper proposed a novel framework that aims to address the data synthesis issue for guardrail model training, especially under dynamic, customized policy, offering a flexible, diverse, and scalable approach in data synthesis
2. The experiments are rigorous; the author repeated their experiments multiple times with different random seeds, with confidence interval reported in their results.

### Weaknesses
1. Some ablation studies are missing. For example, in Table3, the author claimed that the guardrail models trained on BARRED have higher accuracy, claiming that BARRED has good performance. However, this is not surprising, as the fundamental reason why the model is fine-tuned on a dataset generated by BARRED is simply because it's further trained on domain specific dataset, just like a model further fine-tuned on a specific math domain dataset has better performance than a model that is not trained on a domain-specific dataset. What's more important is to show the data quality generated by BARRED, which is somewhat missing in the paper. For example, the author claimed that BARRED addressed a problem that LLM-generated synthetic data usually lacks diversity and have mode collapse issue, but this ablation study is missing throughout the paper. One ablation study that I think is worth adding is to compare the performance with the model fine-tuned on the synthetic data directly generated by LLMs.
2. Diversity evaluation is not very well-grounded. since it's purely llm based evaluation.
3. Some experiment settings is a bit vague. For example, in line 211, how do authors make sure the advocate **never** changes its position?

---

> ### Author Rebuttal · Authors · 2026-03-29
>
> We thank the reviewer for recognizing the novelty of our framework and the rigor of our experiments. We address each concern below, and note that several of these points are addressed by existing ablations in the paper.
>
> ### **Ablation comparing with direct LLM-generated synthetic data**
>
> This ablation is reported in Table 4. The "No verification" baseline generates 1000 synthetic samples using the pipeline without the debate verification or refinement stages. This is equivalent to fine-tuning on synthetic data directly generated by the LLM. The result shows a 27-point accuracy drop on human data (0.58 vs. 0.85), demonstrating that the quality of BARRED's data, not merely domain-specific fine-tuning, drives performance.
>
> ### **Diversity and mode collapse ablation**
>
> This ablation is presented in Figure 3. While Figure 3a provides a complementary LLM-based evaluation of dimension diversity, Figure 3b directly evaluates the downstream impact on accuracy over the human-labeled test sets. Specifically, this evaluation (**which is not LLM-based**) reports prediction accuracy following fine-tuning using dimension instantiation subsets of varying sizes, averaged across five runs with random subset selections to account for variability in dimension importance. The accuracy improves as more dimensions are added, confirming that dimension decomposition systematically addresses mode collapse.
>
> ### **Advocate position mechanism**
>
> As described in Section 3.3, the Advocate "consistently argues for label y using reasoning r, never changing its position." This is by design: we implemented the Advocate to return the same reasoning message (which equals the reasoning output by the generator, Section A.2 in the Appendix) across all debate rounds. It is not an LLM that deliberates or updates its position. This fixed behavior is what makes the debate asymmetric, and is the key architectural distinction from symmetric debate (Du et al., 2023).

---

> > ### Author Rebuttal · Reviewer_pgTe · 2026-04-03
> >
> > Most of my concerns are addressed. Raised score to 4.
> >
> > To AC: this paper may have ethical concerns since it uses a public dataset link in the submission draft. Please double-check. Thanks.

---

### Official Review · Reviewer_QGU3 · 2026-03-12

**Soundness:** 2
**Presentation:** 2
**Significance:** 2
**Originality:** 2
**Overall Recommendation:** 3
**Confidence:** 4

**Summary:**

This paper introduces BARRED (Boundary Alignment Refinement through REflection and Debate), a framework designed to train task-specific guardrail models using purely synthetic data. The pipeline addresses the "diversity" and "faithfulness" challenges of synthetic data through two primary mechanisms: dimension decomposition, which uses verbalized sampling to explore the domain space systematically , and asymmetric multi-agent debate, where a rigid Advocate defends a generated label against a panel of Judges to ensure label fidelity. Experimental results across conversational, agentic, and regulatory tasks demonstrate that small language models (SLMs) fine-tuned on BARRED-generated data can outperform much larger proprietary LLMs and generic guardrail models.

**Compliance With Llm Reviewing Policy:**

Affirmed.

**Key Questions For Authors:**

1. Considering the 10% performance drop on the "Health" human test set compared to the synthetic one, how do you propose to better align the LLM's "imagined" boundary cases with actual human-generated edge cases?

2. How does BARRED perform when using a significantly weaker teacher (e.g., an 8B open-source model)? Does the debate mechanism still yield a student that outperforms the teacher's zero-shot capability, or does the pipeline collapse into "shared hallucinations" when the teacher is less capable?

**Limitations:**

While the authors address technical limitations—including the high resource consumption of iterative LLM calls , the current focus on binary classification, and the inherent risks of label noise or model collapse in synthetic data —across the Results and Analysis, and Conclusion, these points are not centralized into a dedicated section. Additionally, the Impact Statement highlights only positive societal benefits such as safety democratization and reduced human labor, leaving potential negative consequences like the amplification of systemic biases inherited from teacher models or the risk of facilitating automated censorship unaddressed. The authors should consolidate these constraints and provide a more balanced discussion of the societal risks associated with their framework.

**Strengths And Weaknesses:**

**Strengths**

The paper presents a technically sound and highly practical workflow for a critical real-world problem: the need for low-latency, high-accuracy custom policy enforcement. The methodology is well-structured, clearly positioning the "boundary-case" focus as a way to maximize classifier learning efficiency. The empirical results are compelling, particularly the finding that a 3B-parameter model can exceed the performance of frontier-class models like GPT-4.1 when specifically tuned on high-fidelity synthetic data. Furthermore, the qualitative analysis and ablation studies effectively demonstrate the necessity of both dimension decomposition and debate-based verification to prevent model collapse and label noise.

**Weaknesses**

- While the framework generates high-quality "boundary cases," there is a noticeable performance discrepancy between synthetic and human-curated test sets. For instance, in the "Health" task, the model achieves **99% accuracy on synthetic data** but drops to **89% on human data**. This suggests that the "boundary" cases imagined by the LLM generator may not perfectly align with the actual complexities and linguistic nuances of real-world user interactions.

- The generation pipeline is computationally expensive, requiring numerous high-effort LLM calls (reasoning-heavy models like GPT-5-mini) for extraction, sampling, multiple debate rounds, and refinement. Additionally, the system's effectiveness is strictly capped by the inherent capabilities of the teacher LLM; if the generator fails to conceptualize certain violation modes, the resulting guardrail will remain blind to those risks.

- While the combination of techniques is effective, the individual components—verbalized sampling, multi-agent debate, and synthetic boundary-case generation—are largely established in recent literature. The paper functions more as a successful engineering integration than a fundamental theoretical or methodological breakthrough.

---

> ### Author Rebuttal · Authors · 2026-03-29
>
> We thank the reviewer for the positive assessment of our methodology, the recognition of compelling empirical results, and the appreciation of the ablation studies and qualitative analysis. We address each concern below.
>
> ### **Performance gap between synthetic and human test sets**
>
> We note that this gap is not due to insufficient complexity of the generated boundary cases. The debate analysis (Figure 4) shows 30%+ of cases trigger nontrivial deliberation patterns, and the qualitative examples (Tables 2, 5, 6) demonstrate genuinely challenging samples. Rather, the gap stems from the inherent challenge of our task setting: generating comprehensive domain coverage from only a task description and few unlabeled seeds. Figure 3a shows test set coverage saturates around 80%, meaning some real-world variations are not represented in the generation space. Importantly, this coverage gap does not undermine the practical value of the trained models. Our fine-tuned models still outperform all baselines on human data (e.g., on Health: ft-4.1-nano 89% vs. GPT-4.1 85%, GPT-5-mini 73%, OSS-Safeguard-20B 75%). Furthermore, the Spearman rank correlation between the synthetic and human accuracy results across all models is consistently high (ρ = 0.96, 0.99, 0.95, 0.9 for the four tasks, all p < 0.001), confirming that the synthetic benchmark reliably predicts relative model quality on real-world data.
>
> ### **Computational cost and teacher model ceiling**
>
> The one-time cost of building a model per task is on the order of 15 dollars, including sample generation, debate, and fine-tuning. At inference, since the guardrail task is classification (single-token output), cost is dominated by input tokens. Open-source models in the 3B parameter range can be served at ~0.02 dollars/1M input tokens through commercial inference providers, compared to ~2 dollars/1M for frontier proprietary models, representing roughly a 100x cost reduction while achieving higher accuracy. The training cost is thus quickly amortized in production deployment.
>
> Regarding the teacher ceiling: a key insight of our work is that the teacher in BARRED is not a single LLM but an asymmetric debate system combining local reasoning from the generator with an ensemble of independent judges. The ablation directly demonstrates this: when using the same GPT-5-mini as a single-model teacher without verification, accuracy drops to 0.58. The debate mechanism enables the system to exceed the labeling quality of any individual model call, as evidenced by the 30%+ nontrivial debate patterns in Figure 4 where judges correct their previous predictions through deliberation.
>
> ### **Novelty beyond engineering integration**
>
> We would like to highlight three distinct contributions that go beyond integration of existing techniques:
>
> - **Novel task setting.** Training custom guardrails purely from a task description and unlabeled seeds, without any curated safety datasets. Existing guardrail work relies on predefined taxonomies or human-labeled data.
> - **Asymmetric debate mechanism.** The fixed Advocate vs. deliberating Judges design differs fundamentally from symmetric multi-agent debate (Du et al., 2023).
> - **Debate-based validation for synthetic data generation.** Applying debate-based validation to synthetic data generation for downstream classifier training is, to our knowledge, new. The ablation study validates that this is not interchangeable with existing alternatives: Self-Refine, an established refinement method, performs worse than no verification at all (0.53 vs. 0.58), demonstrating that the problem setting itself is challenging and prior methods cannot address it.

---

> > ### Author Rebuttal · Reviewer_QGU3 · 2026-04-03
> >
> > I appreciate the authors' detailed rebuttal. While it addresses some of my technical concerns, my overall assessment of the paper's impact and limitations remains unchanged. I will maintain my original score.

---

### Official Review · Reviewer_QYN1 · 2026-03-12

**Soundness:** 3
**Presentation:** 3
**Significance:** 3
**Originality:** 3
**Overall Recommendation:** 4
**Confidence:** 3

**Summary:**

The paper presents BARRED (Boundary Alignment Refinement through REflection and Debate), a framework for generating synthetic training data to build custom guardrail classifiers. Given only a task description and a small set of unlabeled examples, BARRED produces labeled training data through three key mechanisms: (1) dimension decomposition, which systematically identifies task-relevant axes and their instantiations to ensure diverse coverage of the domain space; (2) boundary-case generation, which targets samples near the decision boundary; and (3) multi-agent asymmetric debate, where a rigid Advocate defends the proposed label against independent Judges to verify label correctness, with iterative refinement for failed samples. The authors evaluate on four tasks across conversational policy enforcement, agentic output verification, and regulatory compliance, showing that small models (3B–14B) fine-tuned on BARRED-generated synthetic data outperform frontier LLMs and dedicated guardrail models. Ablation studies demonstrate that both dimension decomposition and debate verification are essential components.

**Compliance With Llm Reviewing Policy:**

Affirmed.

**Final Justification:**

As showed in the rebuttal acknowledgement, I'd like to maintain my justification from the original review.

**Key Questions For Authors:**

1. Essentially my weakness 3. What happens if you simply have GPT-5-mini classify 1000 unlabeled or randomly generated samples (without dimension decomposition or debate) and fine-tune on those labels?
2. Have you tested BARRED with a weaker or open-source model as the generator?
3. Can you provide precision, recall, and F1 scores alongside accuracy?

**Limitations:**

yes

**Strengths And Weaknesses:**

**Strengths**
1. The idea of generating purely synthetic training data from just a task description and unlabeled examples is compelling and addresses a genuine deployment bottleneck.
2. The pipeline is well-decomposed into interpretable stages (dimension decomposition → boundary generation → debate validation → refinement). Each component has a clear purpose, and the ablations confirm each is necessary. The asymmetric debate design (fixed Advocate vs. deliberating Judges) is a thoughtful departure from standard symmetric debate.
3. Fine-tuned small models consistently outperform much larger LLMs and dedicated guardrail models across all four tasks.
4. The debate path analysis (Figure 4) and the qualitative examples (Tables 2, 5, 6) effectively illustrate how the debate mechanism catches errors and drives meaningful refinement. The example where self-refine performs worse than no verification is a particularly interesting finding.
5. The paper ablates both major components (debate verification in Table 4, dimension decomposition in Figure 3) and provides scaling analysis across model sizes (Figure 2).

**Weaknesses**
1. While four tasks are presented, only two rules from DynaGuard are used (out of 278), and the other two tasks are single benchmarks.
2. The paper uses accuracy as the sole metric, but for guardrail tasks, precision, recall, and F1 are far more informative.
3. The paper does not compare against: (a) direct distillation from GPT-5-mini (i.e., having GPT-5-mini label 1000 samples without the BARRED pipeline), which would isolate the contribution of the framework itself vs. simply using a strong teacher; (b) other synthetic data generation approaches; (c) few-shot fine-tuning with the 10–30 seed examples directly.

---

> ### Author Rebuttal · Authors · 2026-03-29
>
> We thank the reviewer for the thorough and constructive review. We are glad the reviewer found the pipeline well-decomposed, the asymmetric debate design thoughtful, and the self-refine finding interesting. We address each weakness below.
>
> ### **Limited number of tasks/rules evaluated**
>
> The selection of two DynaGuard rules was a deliberate methodological choice. Of 278 rules in the benchmark, the vast majority have fewer than 20 annotated samples or 100% compliance (no violations), making rigorous evaluation impossible. We filtered for rules meeting both criteria (Section 4.1.1), and, additionally, augmented with human annotations to ensure test sets are both large enough and balanced. More importantly, evaluation diversity comes from task types: our four tasks span three fundamentally different domains (conversational, agentic, regulatory) with distinct input modalities and reasoning requirements. Consistent gains across all four provide stronger generalizability evidence than testing many rules within a single domain.
>
> ### **Accuracy as a sole metric**
>
> All test sets are class-balanced (Table 1) with binary labels. We report F1 for the repetition task on human data: GPT-4.1-nano 0.53, GPT-4.1-mini 0.60, GPT-4.1 0.91, GPT-5-mini 0.95, ft-4.1-nano 0.96, ft-Qwen14B 0.93, ft-Qwen3B 0.94. F1 closely tracks accuracy across all models, confirming that accuracy serves as a reliable summary metric with balanced binary labels. We will include full metrics for all datasets in the camera-ready version.
>
> ### **Missing baselines**
>
> We note that Table 4 presents an ablation study designed to isolate exactly these contributions.
>
> **(a) Direct distillation.** The "No verification" ablation is precisely this baseline: GPT-5-mini generates 1000 labeled samples using the same generation pipeline but without debate or refinement. The 27-point accuracy gap on human data (0.58 vs. 0.85) isolates the contribution of the framework itself beyond teacher model strength.
>
> **(b) Other synthetic data methods.** Self-Refine (Madaan et al., 2023) in Table 4 represents an established iterative refinement approach. Its underperformance vs. even no verification (0.53 vs. 0.58) highlights that single-agent critique reinforces errors, while multi-agent debate provides the adversarial pressure needed to catch labeling mistakes. We will consider additional synthetic data baselines for the extended version.
>
> **(c) Few-shot fine-tuning with seeds.** Our seed sets contain only *unlabeled* examples, which are used to capture the expected input structure and characteristics and are not labeled, making them unsuitable for supervised fine-tuning. BARRED's core contribution is precisely bridging from these minimal seeds to labeled datasets and to a fully verified training corpus.
>
> ### **Open-source generator**
>
> Data generation in BARRED requires simultaneously decomposing abstract rules into concrete dimensions, producing realistic, diverse, and challenging samples, and maintaining label correctness through multi-agent debate. This is a substantially more challenging task than classification, and in our experience demands frontier-level instruction following and reasoning capabilities. That said, investigating open-source generators is a valuable direction for future work, particularly as open-source models continue to close the gap on complex generation tasks.

---

> > ### Author Rebuttal · Reviewer_QYN1 · 2026-04-01
> >
> > Thanks author for the rebuttal, as it resolves my concerns, I'd like to maintain my weak accept score.

---

### Decision · Program_Chairs · 2026-04-30

**Decision:**

Accept (regular)

**Comment:**

Scores settle at 4/3/4 after rebuttal.

The paper presents a technically sound framework for synthetic training data generation for custom guardrails, with strong empirical results across diverse tasks. QYN1's and pgTe's concerns about missing baselines and diversity ablations were resolved when the authors pointed to existing experiments. QGU3's maintained weak reject centers on limited novelty as engineering integration, but the consistent outperformance of small fine-tuned models over frontier LLMs across all tasks, combined with the practical deployment value (roughly 100x cost reduction at higher accuracy), represents a solid contribution.